# First Application of a Mixed Porcine–Human Repopulated Bioengineered Liver in a Preclinical Model of Post-Resection Liver Failure

**DOI:** 10.3390/biomedicines12061272

**Published:** 2024-06-07

**Authors:** Philipp Felgendreff, Seyed Mohammad Hosseiniasl, Anna Minshew, Bruce P. Amiot, Silvana Wilken, Boyukkhanim Ahmadzada, Robert C. Huebert, Nidhi Jalan Sakrikar, Noah G. Engles, Peggy Halsten, Kendra Mariakis, John Barry, Shawn Riesgraf, Chris Fecteau, Jeffrey J. Ross, Scott L. Nyberg

**Affiliations:** 1Department of Surgery, Mayo Clinic, Rochester, MN 55902, USA; hosseiniaslm@gmail.com (S.M.H.); minshew.anna@mayo.edu (A.M.); amiot.bruce@mayo.edu (B.P.A.); wilken.silvana@mayo.edu (S.W.); ahmadzada.boyukkhanim@mayo.edu (B.A.); nyberg.scott@mayo.edu (S.L.N.); 2Department of General, Visceral and Transplantation Surgery, Hannover Medical School, 30625 Hannover, Germany; 3Division of Gastroenterology and Hepatology, Mayo Clinic, Rochester, MN 55902, USA; huebert.robert@mayo.edu (R.C.H.); sakrikar.nidhi@mayo.edu (N.J.S.); biomedengineerengles@gmail.com (N.G.E.); 4Miromatrix Medical Inc., Eden Prairie, MN 55344, USA; pnorris@miromatrix.com (P.H.); kmariakis@miromatrix.com (K.M.); jbarry@miromatrix.com (J.B.); sriesgraf@miromatrix.com (S.R.); cfecteau@miromatrix.com (C.F.); jross@miromatrix.com (J.J.R.); 5William J. von Liebig Center for Transplantation and Clinical Regeneration, Mayo Clinic, Rochester, MN 55902, USA

**Keywords:** tissue engineering, acute liver failure, extracorporeal liver assist device

## Abstract

In this study, a mixed porcine–human bioengineered liver (MPH-BEL) was used in a preclinical setup of extracorporeal liver support devices as a treatment for a model of post-resection liver failure (PRLF). The potential for human clinical application is further illustrated by comparing the functional capacity of MPH-BEL grafts as assessed using this porcine PRLF model with fully human (FH-BEL) grafts which were perfused and assessed in vitro. BEL grafts were produced by reseeding liver scaffolds with HUVEC and primary porcine hepatocytes (MPH-BEL) or primary human hepatocytes (FH-BEL). PRLF was induced by performing an 85% liver resection in domestic white pigs and randomized into the following three groups 24 h after resection: standard medical therapy (SMT) alone, SMT + extracorporeal circuit (ECC), and SMT + MPH-BEL. The detoxification and metabolic functions of the MPH-BEL grafts were compared to FH-BEL grafts which were perfused in vitro. During the 24 h treatment interval, INR values normalized within 18 h in the MPH-BEL therapy group and urea synthesis increased as compared to the SMT and SMT + ECC control groups. The MPH-BEL treatment was associated with more rapid decline in hematocrit and platelet count compared to both control groups. Histological analysis demonstrated platelet sequestration in the MPH-BEL grafts, possibly related to immune activation. Significantly higher rates of ammonia clearance and metabolic function were observed in the FH-BEL grafts perfused in vitro than in the MPH-BEL grafts. The MPH-BEL treatment was associated with improved markers of liver function in PRLF. Further improvement in liver function in the BEL grafts was observed by seeding the biomatrix with human hepatocytes. Methods to reduce platelet sequestration within BEL grafts is an area of ongoing research.

## 1. Introduction

Acute liver failure (ALF) is a life-threatening critical illness with an overall mortality rate of 30–50% [1]. Common etiologies include drug-induced liver injury, viral infections or autoimmune disease but up to 50% of cases have no definitive cause [2]. Irrespective of the etiology, liver transplantation remains the only successful treatment for ALF if standard medical therapy (SMT) fails.

The limitations of liver transplantation include the restricted availability of donor organs, the potential for intra- and postoperative complications and the need for lifelong immunosuppression. SMT attempts to stabilize the patient and prevent extrahepatic organ dysfunction. However, SMT alone is often insufficient to support the metabolic, detoxification and synthetic functions of the injured native liver.

One approach to overcome the limitations of SMT is extracorporeal liver support, analogous to hemodialysis for renal failure. The goal of extracorporeal liver support therapy is to provide essential detoxification functions, such as ammonia clearance, and to bridge the ALF patient to spontaneous recovery or liver transplantation.

Two general categories of extracorporeal liver support devices have emerged to achieve this goal: purely artificial devices and those with a biological or cellular component, termed bioartificial liver support devices [3]. Artificial liver devices include Molecular Adsorbent Recirculating Systems [MARS™] and single-pass albumin dialysis (SPAD) [4], which both use a combination of physical or chemical gradients to absorb or eliminate toxins, waste products and pro-inflammatory mediators. Accumulation of these harmful molecules in liver failure are, in part, responsible for the extrahepatic manifestation of ALF, such as cerebral edema, lung injury and acute renal failure. Despite theoretical benefits from artificial liver devices, clinical studies of these devices have yet not shown a survival benefit in ALF patients [4,5,6,7].

Based on these observations, cell-based, bioartificial liver support devices were investigated and developed in experimental settings as a more biological alternative approach to detoxification and metabolic support of the ALF patient [8]. The differences between the designs of these cell-based devices include their source and number of cells, configuration of the cell compartment of the device and whether the patient’s blood has direct or indirect contact with the cells. Indirect contact requires a semipermeable membrane to avoid direct contact between recipient and bioreactor cells if a xenogeneic cell source is used. A semipermeable membrane also facilitates the use of immortalized human cell lines, such as the human hepatoblastoma lines HepG2 and C3A [9], to mitigate the risk of immortalized cells entering the patient’s circulation [10,11]. Due to the unfavorable metabolic profile of immortalized cells, primary hepatocytes are the most promising option for a cell-based bioartificial liver support device. Both primary porcine hepatocytes (PPHCs) and primary human hepatocytes (PHHCs) have been considered for human therapy because of their similar metabolic profiles, including the capacity for ureagenesis. The advantages of PPHCs are their availability and abundant supply, while human hepatocytes avoid xenogeneic concerns.

The success of liver transplantation as a treatment for ALF suggests that bioengineered liver (BEL) grafts may provide another extracorporeal option to support ALF patients. In fact, the process of liver decellularization and recellularization offers a novel and highly promising method to bioengineer liver grafts. The decellularization process yields a natural scaffold for seeding with isolated primary donor cells to produce a bioengineered graft with the desired functionality and immunogenicity.

The new graft is both biocompatible and preserves the three-dimensional architecture required for high cell density and blood flow rates approaching normal portal blood flow. Furthermore, BEL grafts potentially offer an endless source of organs for extracorporeal or implantation therapy.

The functional revascularization of BEL grafts has already been established [12,13]. We first demonstrated continuous hemoperfusion of an implanted BEL for up to 20 days without systemic anticoagulation in a porcine implant model. The BEL grafts were seeded with human umbilical vein endothelial cells (HUVECs) to establish a reendothelialized vascular network [12]. We next developed a method for isolating, culturing and seeding both HUVECs and primary liver cells into second-generation BEL grafts. Our initial studies were conducted using BEL grafts containing mixed populations of porcine hepatocytes and human endothelial cells. These hybrid grafts, termed MPH-BEL grafts, were characterized for liver-specific functions under both in vitro and in vivo conditions [13]. The current study design was intended to characterize the functionality of MPH-BEL grafts implemented in an extracorporeal liver supporting device to treat an established large animal model of ALF. A porcine model of post-resection liver failure (PRLF) was utilized because of its clinical relevance and reproducibility [14]. Furthermore, fully humanized BEL (FH-BEL) grafts were produced using both HUVECs and human hepatocytes. These FH-BELs were assessed in an in vitro model and compared to MPH-BEL grafts to determine the current state of a humanized approach in preparation for phase 1 clinical studies of this novel technology.

## 2. Materials and Methods

### 2.1. Production of Bioengineered Livers (BELs)

#### 2.1.1. Porcine Organ Procurement and Whole Liver Decellularization

Whole porcine livers (500 to 700 g) were procured from slaughterhouse pigs. Following the cannulation of the portal vein (PV), the hepatic artery, the suprahepatic Inferior Vena Cava (sIVC), the infrahepatic Inferior Vena Cava (iIVC) and the common bile duct (BD), the organ was rinsed with phosphate-buffered saline (PBS; Corning 21–040-CMX12) and flushed with saline. Following E-beam treatment (E-Beam Services, Cranbury, NJ, USA), the decellularization process was initiated by perfusing the organ with 1% Triton X-100 (Amresco, Solon, Ohio M143) and 0.6% sodium dodecyl sulfate (Amresco, Solon, Ohio 0227) via PV and sIVC (target pressures: 12 to 17 mmHg). The decellularized livers were disinfected with peracetic acid (PAA; U.S. Water, New Port Richey, FL, USA, BI0032–6) and stored in PBS at 4 °C until the seeding procedure was initiated.

#### 2.1.2. HUVEC Cell Culture

HUVECs (Lonza, Cohasset, MN, USA, C2517A) were cultured at 37 °C and 5% CO_2_ in antibiotic-free endothelial cell growth media (R&D Systems, Minneapolis, MN, USA, CCM027) supplemented with 2% fetal bovine serum (FBS) (Corning, Glendale, Arizona), 50 mg/L ascorbic acid (Sigma, St. Louis, MO, USA), 1 mg/L hydrocortisone (Sigma), 20 μg/L Fibroblast Growth Factor (FGF) (R&D Systems), 5 μg/L Vascular Endothelial Growth Factor (VEGF) (R&D Systems), 5 μg/L Epidermal Growth Factor (EGF) (R&D Systems), 15 μg/L R3 Insulin-like Growth Factor (IGF) (Sigma), 1000 U/L heparin (Sigma) and 1.5 μM acetic acid (Sigma). Cells were harvested with 0.25% trypsin-EDTA (Thermo Fischer, Waltham, MA, USA, 25200056) at 90–100% confluence and prepared for the seeding procedure.

#### 2.1.3. Isolation of Primary Porcine Hepatocytes (PPHCs)

Primary porcine hepatocytes were isolated from fresh whole livers (450–1000 g) harvested from female pigs following the approval of the Institutional Animal Care and Use Committee (IACUC) (No. 040420). The isolation of the PPHCs was performed as previously described by enzymatic digestion using 100 mg of Liberase TM (Sigma, 5401127001) and 5 mM CaCl2. The cell suspension was filtered through an 8″ wide mesh strainer, followed by a series of mesh sieves (250 μm (VWR, Batavia, IL, USA, 57334-466), 125 μm (VWR, 57334-474), 70 μm ((Thermo Fischer, Waltham, MA, USA, NCO446099). After two washing procedures using William’s E + 10% FBS media, the cell viability and yield were quantified by trypan blue dye exclusion on a hemocytometer. The final cell pellet was resuspended in a final volume of 1 L using UW until further use for the seeding of the acellular scaffold.

#### 2.1.4. Isolation of Primary Human Hepatocytes (PHHCs)

Human livers not accepted for transplantation were obtained from Donor Network West, NDRI and Southwest Transplant Alliance. These livers were flushed per standard organ procurement protocol and transported from the site of organ procurement to Eden Prairie, MN, USA by cold static preservation at 4 °C. To isolate primary human hepatocytes, procured livers were first flushed with cold Lactated Ringer’s Solution (LRS; Hutchins & Hutchins, VA, USA) via the hepatic artery and PV to remove preservation solution. The right and left lobes were resected and exposed vessels were cannulated for perfusion. Liver tissue digestion was performed as previously described using both collagenase MA (3 mg/L; Vitacyte) and protease BP (2.5 mg/L; Vitacyte). Digested tissue was diluted in PHHC isolation medium (HHIM; DMEM with 10% FBS) and sieved (1000, 500, 250, 90 µm). Hepatocytes were centrifuged (110× *g* force, room temperature, 10 min) and washed once with HHIM. Hepatocyte viability and yield were quantified on a hemocytometer by trypan blue dye exclusion. The final cell pellet was suspended in an equivalent volume of UW Belzer solution.

#### 2.1.5. Production of Vascularized BEL Scaffolds

Porcine liver scaffolds were prepared by an established technique [12]. Briefly, scaffolds were installed into the bioreactor culture station (Figure 1A) and perfused with Bioreactor Media (Williams’ E based medium supplemented with 1.5% FBS (Corning), 50 mg/L ascorbic acid (Sigma), 1 mg/L hydrocortisone (Sigma), 20 μg/L FGF (R&D Systems), 5 μg/L VEGF (R&D Systems), 5 μg/L EGF (R&D Systems), 15  μg/L R3 IGF (Sigma), 1000 U/L heparin (Sigma), 1.5 μM acetic acid (Sigma), 2 mL/L human insulin (Novolin), 3 g/L human albumin (CSL Behring), 150 μg/L linoleic acid (Sigma), 0.1 μM dexamethasone (Sigma), 40 ug/L human glucagon (Novaplus), 6 mg/L human transferrin (Sigma), 20 µg/L Gly-His-Lys (Sigma), 0.1 μM copper sulfate, 30 nM sodium selenite, 50 pM zinc sulfate, 1 g/L l-carnitine (Sigma), 0.2 g/L l-arginine (Sigma) and 10 mg/L glycine (Sigma)) for 72 h prior to the reendothelialized and hepatocyte seeding procedures.

In the first step of reendothelialization, HUVECs (1.6 × 10^8^ cells) were introduced into the liver by retrograde perfusion of the sIVC at 300 mL/min. After first introduction, perfusion of the sIVC was discontinued and scaffolds underwent 1 h of static culture to facilitate cell attachment. A second dose of 4.0 × 10^7^ HUVECs was introduced into the scaffolds via continuous perfusion of the sIVC at 300 mL/min. Scaffolds were static cultured overnight. The following day, scaffolds were seeded with a third and fourth dose of HUVECs via the PV perfused with a similar method to that used with the sIVC. Following reendothelialization, vascularized BEL grafts were perfused using Bioreactor Media for up to 19 days with a maximum perfusion flow of 300 mL/min and maximum pressure of 30 mmHg.

#### 2.1.6. Production of MPH-BEL and FH-BEL Grafts

MPH-BEL grafts and FH-BEL grafts were produced from vascularized BEL grafts following day 19 of reendothelialization as described above. Either PPHCs or PHHCs were used to produce MPH-BEL grafts and FH-BEL grafts, respectively. On day 19 of reendothelialization, primary hepatocytes were perfused into the IVC of the vascularized BEL grafts at an infusion rate of 350 mL/min. MPH-BEL grafts were either inoculated with 10 × 10^9^ PPHCs for the preclinical study or 5 × 10^9^ PPHCs for the in vitro study. FH-BEL grafts used in the in vitro study were seeded with 5 × 10^9^ PHHCs.

Prior to data collection, seeded BEL grafts were perfused continuously with Co-culture Media via the PV at a maximum flow rate of 300 mL/min and maximum pressures of 30 mmHg (Figure 1B). BEL grafts were flushed with cold UW during for transport from the bioreactor culture station to the extracorporeal system.

### 2.2. Porcine Model of Post-Resection Liver Failure (PRLF)

Fifteen female domestic pigs (26–30 kg) served as PRLF animals in a randomized trial of extracorporeal BEL treatment; 10 blood type-matched male domestic pigs (50–60 kg) served as blood donation animals. All animals were obtained from a local vendor (Manthei Hog Farm, Elk River, MN 55330, USA).

As outlined in Figure 2A, all female pigs underwent placement of an ambulatory intracranial pressure probe and double-lumen venous catheter under general anesthesia three to four days prior to 85% liver resection surgery. On the day of 85% liver resection, to induce PRLF, all animals underwent the following sequence of events: blood draw and intracranial pressure (ICP) measurement (Tbaseline), pre-hepatectomy CT scan, 85% hepatectomy and post-hepatectomy CT scan. The completion of 85% liver resection was defined as T0 = 0 h. Animals surviving at least 24 h following hepatectomy (T24 h) were enrolled and randomized into one of three treatment groups, the standard medical therapy alone (SMT) control group, SMT + extracorporeal circuit (SMT + ECC) control group and SMT + MPH-BEL treatment group. All PRLF pigs underwent continuous monitoring for up to 24 h after initiation of the group’s respective therapy, or until reaching one of the predefined humane endpoints of the study.

#### 2.2.1. Anesthesia and Surgical Procedures

All animal husbandry and procedures were conducted in accordance with the guidelines set forth by the Mayo Foundation Animal Care and Use Committee. The surgical procedures were performed under general anesthesia induced with an intramuscular injection of Telazol (5 mg/kg) plus Xylazine (2 mg/kg) and maintained with 1–3% inhaled isoflurane. During therapy, the animals in each group were sedated with 0.1–0.2 mg/kg/min propofol.

#### 2.2.2. Placement of Intracranial Pressure Probe and Double Lumen Central Venous Catheter

The intracranial pressure probe and the double lumen venous catheter placement were performed in a single session as previously described [14]. All pigs were placed in prone position, a 3 cm diameter semi-circular skin incision was made on the skull just above the level of the eyes. After exposing the skull, a 5 French (~4 mm) burr hole was drilled through the skull. The tip of the implantable intracranial pressure probe was placed directly in the brain tissue and its transmitter was positioned in the subcutaneous space. The incision was closed over the intracranial pressure probe. Following placement of the intracranial pressure probe, a cuffed central venous catheter (CVC) was inserted into the right jugular vein under ultrasound guidance and Seldinger technique. Catheter position in the right atrium was confirmed by fluoroscopy. The catheter was tunneled to the pig’s back where it exited and was secured with catheter cuff at the level of the scapula.

#### 2.2.3. The 85% Hepatectomy

Under general anesthesia and in a supine position on the operation table, resection pigs underwent the 85% hepatectomy according to the description of Chen et al. [14]. Briefly, the left and right lateral liver lobes, the PV, hepatic artery and hepatic veins were identified and isolated through a midline incision. The left and middle lobe PVs were isolated and ligated with 2.0 silk to minimize blood loss during resection. Hepatectomy proceeded from left to right as follows: removal of the left lateral lobe, both left and right medial lobes and most of the right lateral lobe; the caudate lobe and small part of the right lateral lobe were left as 15% remnant liver parenchyma. The parenchymal transection was completed using a clamp and crush technique. Following resection, the abdominal wall fascia and the abdominal wall and skin were closed in layers. The extent of hepatectomy and percent resection were confirmed by CT volumetry comparing the native (pre-resection) liver volume with the remnant (post-resection) liver volume.

#### 2.2.4. Standard Medical Therapy (SMT) after 85% Hepatectomy

All hepatectomy animals were transferred to the post-anesthesia recovery room for the first 24 h and received continuous IV 5% dextrose normal saline (D5NS) to maintain a physiological blood glucose level. During the entire recovery period, blood glucose and ICP measurements were performed at 2 h intervals. Blood glucose less than 75 mg/dL was treated with 5 mL of 50% dextrose. At 6 h intervals, additional blood samples (aspartate amino transferase (AST), international normalized ratio (INR), total bilirubin, blood glucose, electrolytes, Hb, platelet count and ammonia concentration) were taken from the central line.

Twenty-four hours after surgery (T24 h), all animals underwent the following sequence of events prior to transfer to the operation table: induction of sedation by using propofol infusion at 80 mcg/Kg/min, intubation and urine catheter placement. Blood pressure, heart rate, oxygen saturation and ICP were monitored continuously throughout the therapy period. Intravenous fluid including D5NS and Plasmalyte were given together at the minimum rate of 100 mL/h to the maximum rate of 400 mL/h to maintain mean arterial pressure (MAP) above 50 mmHg and blood glucose above 75 mg/dL. Dobutamine (2–10 mcg/kg/min) and Phenylephrine (0.5–3 mg/kg/min) were used when the MAP dropped below 50 mmHg despite receiving the maximum amount of IV fluid. Mechanical ventilation was used if the end-tidal PCO_2_ was more than 50 mmHg or oxygen saturation was less than 92% despite applying oxygen.

#### 2.2.5. Extracorporeal Therapy

Resected pigs that were randomized to the SMT + ECC control group and SMT + MPH-BEL treatment group underwent extracorporeal therapy using the two-lumen CVC catheter starting 24 h after 85% hepatectomy. Figure 3 illustrates extracorporeal BEL therapy in a human patient. Prior to the start of therapy, all animals received a 100 mL/kg plasmolyte i.v. bolus to avoid hypovolemia. While the pig was being prepared, the extracorporeal circuit (Baxter’s PrisMax system) was first flushed with 1000 mL plasmolyte and then primed with up to 900 mL of fresh blood harvested from a blood-type match donor pig. Heparin and calcium citrate were used for anti-coagulation during therapy; these agents were administered intravenously to achieve blood-activated clotting time (ACT) between 175 and 225 s and normal ionized calcium level, respectively. ACT was measured hourly during extracorporeal therapy. In the SMT + ECC control group, this setup was used to treat the 85% hepatectomy pig for up to 24 h after liver surgery. The animals included in the SMT + MPH-BEL treatment group were treated with the MPB-BEL by connecting the graft via the portal vein with the extracorporeal circuit. The graft was perfused under physiological conditions (PV flowrate: 300 mL/min, PV pressure: 30 mmHg) during the entire treatment period.

#### 2.2.6. Blood Donation

Under general anesthesia, a midline laparotomy was performed in the blood donor pigs and the infrarenal aorta was exposed. Following mobilization and dissection of the vessel, a 10 G needle connecting to a blood storage bag was inserted in the aorta and 900 mL to 1500 mL blood was collected. The operation was terminated by the euthanasia of the animals by exsanguination. The blood bags were stored at 4 °C until further use in the extracorporeal perfusion circuit.

#### 2.2.7. Study Endpoints

The following predefined humane endpoints were used to terminate the investigation:Survival to 48 h after hepatectomyBlood loss of more than 500 mL during liver resectionSigns of hepatic encephalopathy stage III or IVElevated ICP (>20 mmHg) for over 1 h

Mean arterial blood pressure < 30 mmHg × 60 min at maximum vasopressor support (Plasmalyte flow rate: 400 mL/h, Dobutamine 10 mcg/kg/min, Phenylephrine 3 mg/kg/min).

#### 2.2.8. Animals Were Euthanized by Sodium Pentobarbital Overdose after Reaching One of These Endpoints

##### Histology

Liver tissue was collected from the middle lobe and the remnant liver lobe at the time of liver resection and at euthanization. Samples of BEL graft tissue were obtained from all liver lobes at the end of extracorporeal therapy. All liver tissue was fixed in 5% buffered formalin for 48 h and thin sectioned (5 μm) after paraffin embedding. Thin sections were stained by hematoxylin/eosin (H/E) and Sirius red. Immunohistochemical (IHC) staining was performed on formalin-fixed paraffin-embedded (FFPE) tissues, which were sectioned at 4 µm and placed on Superfrost plus (VWR) slides. Ki67, Cleaved Caspase-3 and fumarylacetoacetate hydrolase (FAH) were used for IHC staining using the following protocols: all sections were digitalized using a MoticEasyScan Pro slide scanner (Motic Digital pathology, San Francisco, CA, USA). Tissue images were analyzed at 4-, 10-, 20- and 40× magnification using imageJ according to known evaluation recommendations [15].

##### Immunofluorescence

Sections of FFPE tissue were deparaffinized by using xylene and decreasing concentrations of ethanol. Antigen retrieval was performed by incubating in 10 mM sodium citrate solution. Quenching was performed using Image-it FX signal enhancer (Invitrogen, Carlsbad, CA, USA). The slides were incubated overnight with the primary antibodies (rabbit anti-CD61 1:100 dilution (Invitrogen) and goat anti-LYVE1 1:200 dilution (R&D Systems)). Donkey anti-goat IgG AlexaFluor™ 488 and donkey anti-rabbit IgG AlexaFluor™ 546 (Invitrogen) were used as secondary antibodies in a 1:250 dilution. Following one hour of incubation with the secondary antibodies, the Prolong Gold antifade reagent (with DAPI) (Invitrogen) was applied to the slides and a cover slip was placed. The slides were digitalized using the EVOS™ M7000 imaging system (Invitrogen).

##### Interleukin-6

Custom panels were created using available porcine Luminex magnetic bead assays from Biotechne R&D Systems (Minneapolis, MN, USA). Samples were analyzed in duplicate after a 1:2 dilution in calibrator diluent, according to the manufacturer’s protocol. Fluorescent signals were captured and quantified with a Luminex 200 instrument using settings suggested by the kit protocol.

### 2.3. In Vitro Perfusion of MPH-BEL and FH-BEL Grafts

HUVEC-BEL scaffolds seeded with either porcine hepatocytes (MPH-BEL) or human hepatocytes (FH-BEL) were used for long-term in vitro perfusion studies. These grafts were perfused with Bioreactor Medium through the PV at a rate of 300 mL/min and a pressure of up to 30 mmHg for up to three days. Media changes were conducted daily unless otherwise described. Grafts were sampled for histological analysis after in vitro perfusion.

#### 2.3.1. Analysis of Metabolites

Media samples were collected from bioreactors daily and immediately assayed on a CEDEX Bio HT bioanalyzer (Roche, Boston, MA, USA) to determine glucose and ammonia levels. Additional media samples were collected daily for quantification of urea, alpha-1 antitrypsin (A1AT) and fibrinogen; these samples were stored at −80 °C prior to analysis. Levels of urea (BioAssay Systems, Hayward, CA, USA, DIUR-100), A1AT (Abcam, Waltham, MA, USA; ab189579) and fibrinogen (Abcam, ab241383) were determined by commercial kits.

#### 2.3.2. Ammonia Clearance by BELs

Rates of ammonia clearance by BEL grafts were determined using fresh medium spiked with 200 µM ammonium chloride. Fresh media were exchanged within the bioreactors 14–24 h after hepatocyte seeding. Fresh media were circulated through the grafts for 10 min prior to determining rates of ammonium clearance in graft effluent over the next 60 min. Ammonia concentration was determined by CEDEX Bio HT Bioanalyzer (Roche, Boston, MA, USA).

#### 2.3.3. Statistical Analysis

Data are presented as mean values and standard deviations of mean. The Kruskal–Wallis test in combination with a post hoc test was used to analyze the preclinical in vivo results. The animal survival was shown as Kaplan–Meier curves with the log-rank test. In the case of pairwise comparisons at single study time points, Student’s *t*-test was used. *p*-value < 0.05 was determined significant.

## 3. Results

### 3.1. Animal Characteristics and Operative Variables

The characteristics and operative variables of the PRLF animals are summarized by group in Table 1. No differences in animal weight, pre-hepatectomy liver volume, post-hepatectomy or estimated blood loss were observed between the SMT + MPH-BEL treatment group and either of the two control groups (SMT, SMT + ECC). The mean percentage of liver resection was similar between all three groups (SMT, 88.0%; SMT + ECC, 86.8%; SMT + MPH-BEL, 87.5%).

### 3.2. Survival

Fifteen animals reached the T24 h mark and were enrolled in the study groups. Three animals of the SMT + MPH-BEL group were excluded from further analysis due to a serious adverse event experienced during extracorporeal perfusion (1—heparin overdose associated with intraperitoneal hemorrhage, 2—human error causing air embolism, 3—hyperkalemia from insufficient flush of UW solution from graft). Median survival duration was determined for the remaining 12 animals as follows: SMT (*n* = 5) 43.6 h ± 3.2, SMT + ECC control group (n = 3) 44.0 h ± 6.9 and SMT + MPH-BEL treatment group (n = 4) 47.0 h ± 2.0. The survival duration of the groups was similar based on Kaplan–Meier analysis (*p* = 0.10) (Figure 4).

One of the five animals in the SMT control group survived to the 48 h endpoint. However, the other four SMT control animals (80% of the total) experienced a serious neurological manifestation of PRLF prior to 48 h, thus confirming the reproducibility of the 85% resection model. Three of the four animals that reached a death equivalent neurological endpoint had elevated ICP above 20 mmHg for more than 1 h. These three animals also experienced rapid limb paddling, decerebrate posturing, dilated fixed pupils and non-responsiveness to painful stimuli consistent with cerebral edema and impending brain herniation. According to the predefined humane endpoints of the study, these three animals were euthanized at 40.3 h, 41.3 h and 42.5 h after the hepatectomy. The fourth SMT control animal was euthanized at 45 h after developing grade 4 hepatic encephalopathy.

Two of the three animals in the SMT + ECC group also survived to the 48 h endpoint. The third SMT + ECC control animal was euthanized due to hemodynamic instability (MAP < 30 mmHg) under maximum cardiopulmonary support at 36.2 h after the 85% liver resection. No evidence of sepsis or internal or external hemorrhage were noted during treatment or at necropsy. A severe systemic inflammatory response of PRLF could not be ruled out as the cause of this animal’s hemodynamic instability.

In the SMT + MPH-BEL group, only one animal died due to grade 4 hepatic encephalopathy at 44.2 h after the 85% liver resection. The remaining three animals completed the entire 24 h treatment period and were weaned successfully from the supportive therapy (vasopressor and fluid support) as well as mechanical ventilation. These three animals were observed for a short period after successful weaning (up to 53 h after the extended liver resection) before being euthanized.

### 3.3. Evaluation of the MPH-BEL Functionality in the PRLF Model

For evaluation of the extracorporeal system functionality in the 85% liver resection model, multiple liver-specific parameters were measured. The focus of these tests was to evaluate the detoxification and synthetic function of the PPLC-BEL.

### 3.4. Detoxification Function and Correlating ICP with Ammonia Levels of the MPH-BEL in the PRLF Model

The detoxification capacity of the MPH-BEL was measured by comparing the serum ammonia level (Figure 5A) and the urea production (Figure 5B) in the three study groups. All the study animals showed comparable serum ammonia level prior to the 85% hepatectomy (SMT-TBaseline: 20 ± 14.1 µmol/L; SMT + ECC-TBaseline: 20.5 ± 2.1 µmol/L; SMT + MPH-BEL, 12.3 ± 4.0 µmol/L; *p* > 0.05). Following the 85% hepatectomy, the ammonia levels increased significantly (*p* ≤ 0.05) at the time of randomization into the subsequent treatment groups (SMT-T24: 156.0 ± 61.8 µmol/L; SMT + ECC-T24: 136.5 ± 23.3 µmol/L; SMT + MPH-BEL-T24: 107.3.3 ± 27.0 µmol/L; *p* < 0.05 in all the treatment groups, indicating implementation of PRLF). During the treatment interval, the ammonia level in the respective study groups was not significant different in the respective groups. However, we noticed differences in the time to peak ammonia levels in the individual groups. The average peak ammonia level in the SMT group (203.4 ± 194.6 µmol/L) and the SMT + ECC group (291.5 ± 234.1 µmol/L) were observed after 8.0 h ± 3.5 and 14.4 h ± 8, respectively. The average peak ammonia level in the SMT + MPH-BEL group (288.34 ± 157.3 µmol/L) was observed after 21.0 h ± 3.5 of treatment. The time to peak ICP measurements (peak ICP: SMT: 12.0 h ± 6; SMT + ECC: 14.3 h ± 2.9; SMT + MPH-BEL: 18.0 h ± 6.0) followed a similar pattern (Appendix A).

Additionally, MPH treatment was also associated with significantly greater urea synthesis as compared to the SMT treatment alone in the second half of the treatment period (urea value SMT + MPH-BEL-T42: 11.5 mg/dL ± 1.7; urea value SMT-T42: 8.8 mg/dL ± 1.3; *p*-value: 0.04; urea value SMT + MPH-BEL-T48: 13.3 mg/dL ± 2.9; urea value SMT-T48: 8.6 mg/dL ± 1.5; *p*-value: 0.01).

### 3.5. Synthetic Function of the MPH-BEL in the PRLF Model

INR values (Figure 6), indicating coagulation function, were comparable at the Tbaseline in all the study animals (SMT 1.23 ± 0.1; SMT + ECC 1.3 ± 0.1; SMT + MPH-BEL 1.28 ± 0.15; *p* > 0.05). There was a significant increase in INR 24 h after the hepatectomy in all the treatment groups (Tbaseline vs. T24; SMT *p* = 0.001 SMT + ECC *p* = 0.003, SMT + MPH-BEL: *p* = 0.006), indicating sufficient PRLF induction in all the study animals.

During the treatment period, INR in the SMT group and the SMT + ECC group remained above the cut-off for acute liver failure of 1.5 recommended by the American Association for the Study of Liver Disease and the European association for the study of the liver [16,17]. However, in the SMT + MPH-BEL group, INR decreased to significantly below this value after 18 h of treatment (T24 1.78 ± 0.191 vs. T42 1.48 ± 0.10 *p* = 0.03). Changes in the other synthesis parameters, such as albumin syntheses or the degree of liver injury (AST, ALT), between all three groups at all the time points were not observed Appendix A.

### 3.6. Hematological Markers during MPH-BEL Treatment

All the animals included in the study showed comparable CBC and IL6 results (Figure 7) at the baseline and at the randomization (T24) of the study. Following the randomization, both HCT and platelet counts were stable in the SMT group over the treatment period. After the randomization, both HCT and platelet counts in the SMT group remained mostly stable throughout the following treatment period. In both the SMT + ECC group and the SMT + MPH-BEL, a decrease in HCT and platelets counts was observed. However, HCT and platelet counts both decreased significantly more during the SMT + MPH-BEL therapy in comparison to the SMT or the SMT + ECC group. In the SMT + MPH-BEL group, the average HCT decreased from 30.0% ± 3.6 at T24 to 14.3% ± 1.8 at the end of the treatment interval (*p* < 0.05). Similarly, the SMT + MPH-BEL treatment led to a significant decrease in platelet counts starting immediately after initiating therapy (platelet count SMT + MPH-BEL-T24: 234.8 × 10^6^ ± 53.3 × 10^6^; platelet count SMT + MPH-BEL-T30: 20.0 × 10^6^ ± 18.1; *p* = 0.002). The lowest platelet counts in this group were observed at T48 (4.3 × 10^6^ ± 3.8).

Interestingly, the white blood count also decreased significantly in the MPH-BEL group compared to the SMT and SMT + ECC groups (Figure 7C).

However, the IL6 concentrations as an additional marker of immune responsiveness during the SMT + MP-BEL treatment did not indicate an inflammatory reaction associated with the graft or the extracorporeal system.

### 3.7. Histology of BEL Grafts

A histological comparison of the liver samples after the in vitro perfusion study and PRFL treatment is shown in Figure 8. Viable hepatocytes were identified in both groups (black arrows). However, hemoperfusion in the post-PRLF group resulted in an infiltration of immune cells and platelet sequestration in the BEL grafts. This finding was confirmed by CD 61 staining as well as by immunofluorescence staining showing an accumulation of platelets near the extracellular matrix in the post-treatment samples. The extracellular matrix was not affected by the treatment and remained intact after the hemoperfusion.

### 3.8. In Vitro Comparison of MPH- BEL and FH-BEL Grafts

In vitro analysis of the MPH-BEL and FH-BEL grafts showed significant differences in the short- and long-term perfusion periods in regard to the synthetic and detoxification functions of the different grafts (Figure 9).

A higher capacity for ammonia clearance was observed in the MPH-BEL bioreactors on the first day of in vitro perfusion compared to the FH-BEL bioreactors. However, the urea genic capacity of the MPH-BEL bioreactor declined more rapidly over the long-term perfusion period. On day two, both groups showed similar ammonia clearances. On the third day, ammonia clearance by the MPH-BEL was undetectable and was associated with ammonia accumulation in the perfusion circuit. By contrast, the ammonia clearance of the FH-BEL remained relatively stable through the third day of the experiment. Additionally, the FH-BEL showed a greater capacity for A1AT and fibrinogen synthesis as compared to the MPH-BEL with long-term perfusion. Ammonia detoxification paralleled urea production in the FH-BEL.

## 4. Discussion

This preclinical large animal study demonstrates the potential utility of an extracorporeal perfusion system using MPH-BEL grafts as a treatment for PRLF. The results of the study show that MPH-BEL treatment leads to improved detoxification and synthetic function as indicated by the extended time of both the peak ammonia level and peak ICP results as well as accelerated INR normalization in the PRLF model. Furthermore, the preclinical testing of the MPH-BEL also identified important aspects for the further optimization of BEL grafts on the pathway to clinical application.

This study represents a step in the development and FDA approval process of the BEL and its role as an extracorporeal liver support device and eventually as a whole implantable organ. To test the functionality of the BEL graft, an 85% porcine liver resection model was used to induce PRLF in our first set of in vivo studies, followed by further evaluation under in vitro perfusion conditions. Both models have been shown to be highly reproducible in previous studies [12,13,18]. The porcine PRLF model has a strong correlation with post-resection ALF in humans in terms of pathophysiological pathways and clinical signs. Reproducibility of this model was high in terms of percent liver resection by CT volumetric measurements, estimated blood loss and manifestations of ALF in each of the three study groups. Thus, this model allows for the quantification of the detoxification and synthetic functions of the BEL graft in a preclinical setting of PRFL. In combination with the already demonstrated maintenance of physiological blood flow in a BEL graft after heterotopic or orthotropic transplantation, this sets the stage for potential trials [12,13,18].

The study was able to show early improvements in the synthesis of coagulation factor (INR) in combination with improved urea synthesis in the SMT + MPH-treated groups as compared to both control groups. The synthesis of urea from ammonia represents an important parameter of hepatocyte function and is frequently used in 2-D and 3-D cell culture systems as a marker for hepatocyte-specific function [19,20]. The more remarkable synthesis of urea in the SMT + MPH-BEL group can, therefore, be interpreted as a confirmation of improved hepatocyte function in the BEL graft and suggests a potential role for bioartificial liver support device treatment in preventing the development of cerebral edema.

The need to lower ammonia levels in the blood of patients with ALF is well established, as ammonia levels correlate with astrocyte swelling, brain edema and subsequent brainstem herniation, leading to brain death in these patients [21,22,23]. Minor differences in ammonia clearance in the MPH-BEL group compared to both control groups may be a limitation of the MPH-BEL grafts. However, the delayed peak ammonia levels in the MPH-BEL group are consistent with some therapeutic benefits of BEL grafts. Considering both the previously reported high rates of ammonia clearance in vitro [12] and these in vivo results, it becomes apparent that by further optimizing the BEL, clinical application of this approach become feasible [14].

Several dozen bioartificial liver support devices have been previously described under in vitro and extracorporeal conditions. However, only a small subset of these liver support systems was shown to reduce blood ammonia levels during preclinical ALF treatment [14,24]. Taking into consideration all of the previous experiences, it appears there is a minimum hepatocyte mass required for adequate detoxification to allow stabilization and recovery of the ALF patient. In vivo repopulation studies have shown that a minimum of 2 × 10^8^ cells/kg body weight were approximately needed to replace liver function in recipients with ALF [25]. Based on this finding, we hypothesized that 5 × 10^9^ porcine hepatocytes in the 30 kg pigs would be sufficient to rescue the recipient from PRLF.

From in vivo cell transplantation studies, which admittedly are entirely different from the methods used in the production of these BELs, it is known that only 20% of the applied cells were engrafted in the respective organ [25]. However, whether in vivo repopulation studies are likely not at all comparable to the reseeding of BEL organs is uncertain [26]. Due to the physiological blood flow and conditions in in vivo cell transplant studies [27], the cellular infusion conditions can only be manipulated to a minor extent to enhance the number of engrafted cells. In contrast, the repopulation conditions of BEL grafts are well monitored and optimized. A combination of static and dynamic perfusion of BEL grafts during the decellularization procedure can be easily achieved and maintained. Multiple studies have already confirmed the sufficient engraftment of hepatocytes using this setup [27,28,29]. However, the specific number of cells engrafted in the organ during the reseeding procedure is unknown. Due to the controlled conditions of the repopulation procedure in the BEL organs, we expect a significantly higher rate of cells engrafted into the cell-free scaffold during the repopulation procedure. Independent of this assumption, further optimization of the repopulation procedure leading to a higher engraftment and expansion of hepatocytes is required to increase the functional capacity of the graft. Moreover, we have previously reported mitotic activity by engrafted hepatocytes in BEL grafts, though the extent to which regeneration is possible in these tissue engineered grafts requires further study and characterization.

In addition to enhancing the engraftment of functional hepatocytes, the study showed that using FH-BELs significantly increased the detoxification and synthesis capacity of the grafts. Especially with longer-term treatment intervals of several days prior to liver transplantation procedure [30], our data indicated a benefit of fully humanized BELs in comparison to mixed porcine–human BELs. The availability of human hepatocytes represents a potential limitation to this approach. Novel methods of hepatocyte cryopreservation continue to be pursued.

Along with the putative need to increase hepatocyte mass in BEL grafts, the study also identified the need to mitigate clot formation in MPH-BEL grafts as an area of future research. Clot formation and platelet deposition in the grafts needs to be interpreted mechanistically in the context of the near-complete reendothelization of the grafts. One potential explanation for these changes could be the reduced mechanical strain [31] of BEL grafts in combination with the physiological portal venous flow, leading to an activation of the coagulation cascade. By hemoperfusing the grafts at a rate greater than 200 mL/min, the established HUVEC cell layer may be partially disrupted and may expose the extracellular matrix to platelet deposition. Considering previous publications [12,13], it should be noted that the grafts lined with HUVECs alone remained patent and did not experience intra-graft clot formation, suggesting the presence of hepatocytes, their antigens and their protein products as contributing factors to clot formation in the MPH-BEL grafts.

In conclusion, the study represents the first report of mixed porcine and humanized BEL grafts in a preclinical model of PRLF. Functionality of the MPH graft with regards to synthetic activity and improved liver homeostasis was established.

Further optimization is required to increase the number of vital hepatocytes in BELs, which will in turn lead to improved liver-specific function of the graft and associated survival benefits in the clinical setting. The new in vitro data for FH-BELs are especially promising in this regard and warrant further investigations with the goal of making extracorporeal, and eventually fully implantable BEL grafts, a clinical reality. The enormous clinical potential of the BEL was further also noted by the Food and Drug Administration, which has approved this approach for clinical use with an Investigational New Drug Application.

## Figures and Tables

**Figure 1 biomedicines-12-01272-f001:**
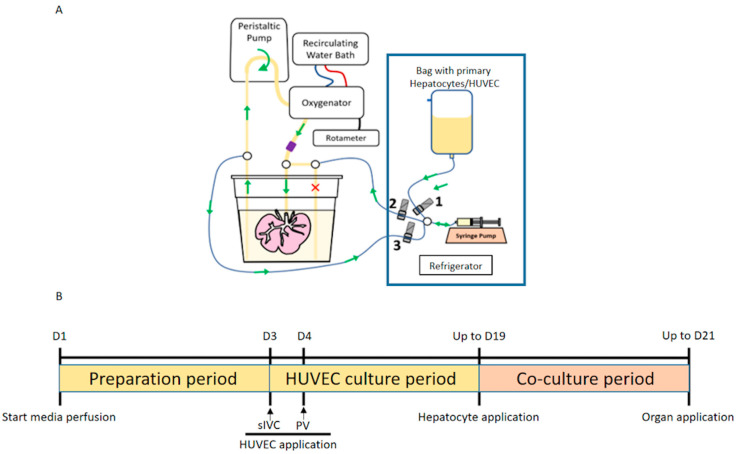
Reseeding setup and timeline. (**A**) Automated perfusion system used for reendothelialization and repopulation of an acellular scaffold; 1–3 are marking the automated clamps controlling the repopulation procedure (**B**) reendothelialization and repopulation timeline.

**Figure 2 biomedicines-12-01272-f002:**
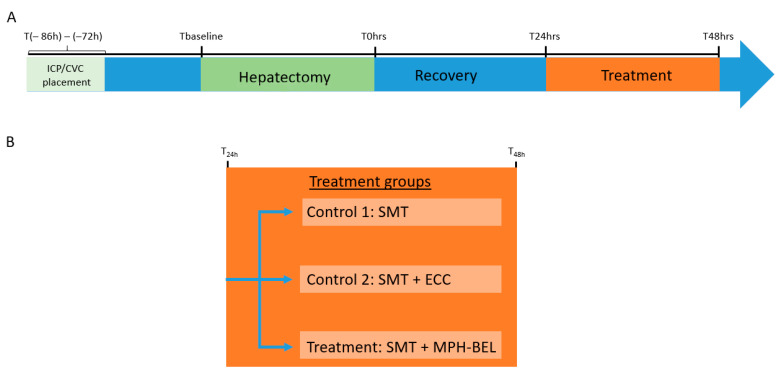
Study timeline (**A**) and treatment groups (**B**). SMT: standard medical therapy, ECC: extracorporeal circuit without BEL graft, MPH-BEL: Bioengineered livers containing mixed populations of porcine hepatocytes and human endothelial cells.

**Figure 3 biomedicines-12-01272-f003:**
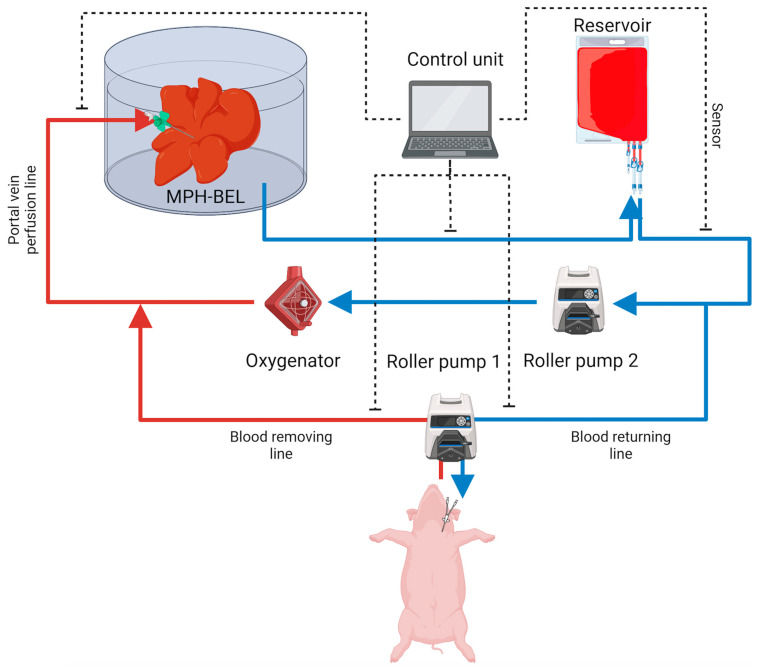
Illustration of Extracorporeal BEL treatment in the PRLF model including the BEL graft, the control unit, roller pumps and the reservoir.

**Figure 4 biomedicines-12-01272-f004:**
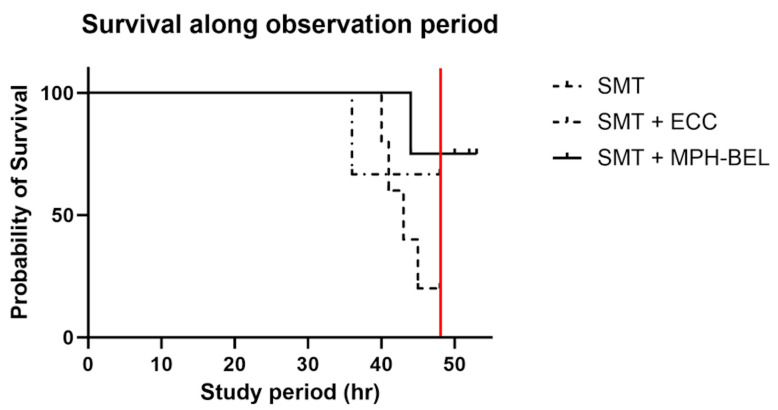
Kaplan–Meier curve comparing the survival rate of the three investigated study groups using the log-rank test. SMT: standard medical therapy, SMT + ECC (extracorporeal circuit without BEL graft), SMT + MPH-BEL (bioengineered livers containing mixed populations of porcine hepatocytes and human endothelial cells). The red line marks 48 h, the end of treatment interval. (*p* = 0.10).

**Figure 5 biomedicines-12-01272-f005:**
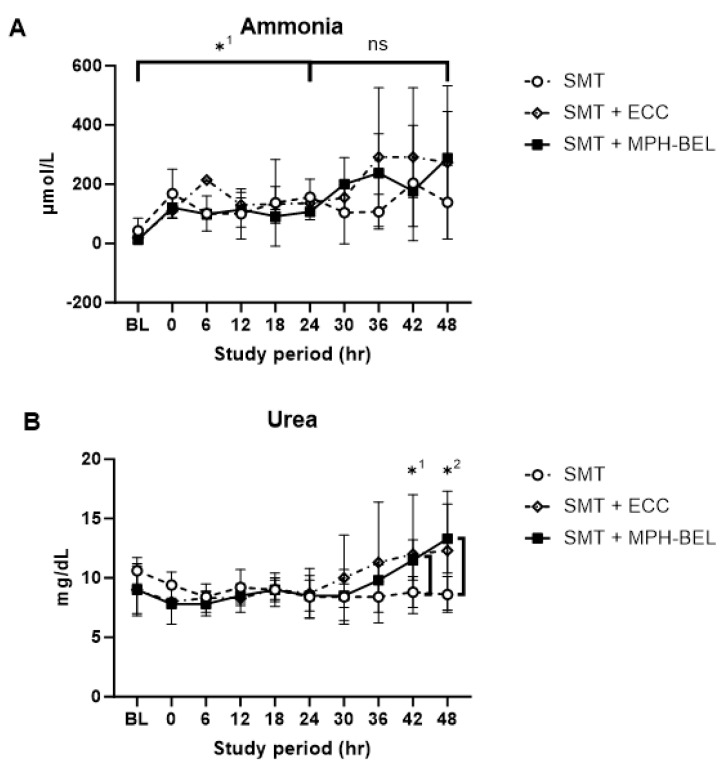
Mean serum concentration of ammonia (**A**) and urea (**B**) are presented as a function of study period for each treatment group. SMT: standard medical therapy is indicated by open circles, SMT + ECC: extracorporeal circuit without BEL grafts is indicated by open diamonds, SMT + MPH-BEL: bioengineered livers containing mixed populations of porcine hepatocytes and human endothelial cells is indicated by closed squares. (**A**) *^1^ Significant difference (Student’s *t*-test; *p* < 0.05) in ammonia level in the comparison of time point baseline (Tbaseline) and 24 h after 85% liver resection (T24) in all study groups, ns: not significant (**B**) *^1,^*^2^ Significant difference (Student’s *t*-test; *p* < 0.05) of urea level in the comparison of SMT and SMT + MPH-BEL at 42 h (T42) and 48 h (T48) after 85% liver resection.

**Figure 6 biomedicines-12-01272-f006:**
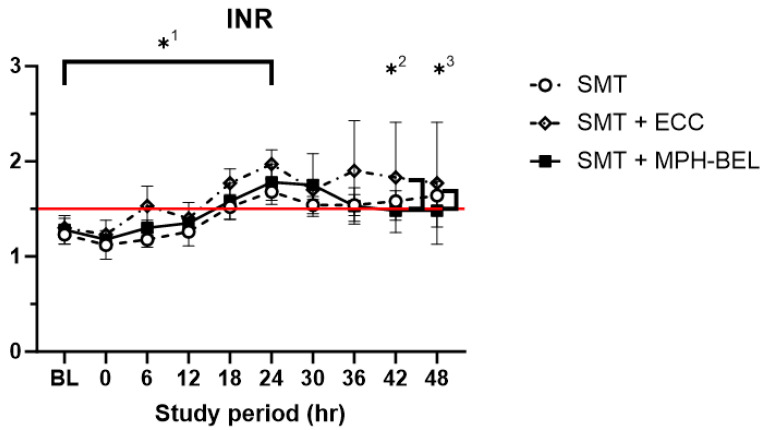
INR values during the study period. Red line indicates a cut-off of 1.5; SMT: standard medical therapy, SMT + ECC (extracorporeal circuit without BEL graft), SMT + MPH-BEL (bioengineered livers containing mixed populations of porcine hepatocytes and human endothelial cells). *^1^ Significant difference (Student’s *t*-test; *p* < 0.05) of INR level in the comparison of time point baseline (Tbaseline) and 24 h after 85% liver resection (T24) in all study groups; *^2,^*^3^ Significant difference (Student’s *t*-test; *p* < 0.05) of INR level in the comparison of SMT and SMT + MPH-BEL at 42 h (T42) and 48 h (T48) after 85% liver resection.

**Figure 7 biomedicines-12-01272-f007:**
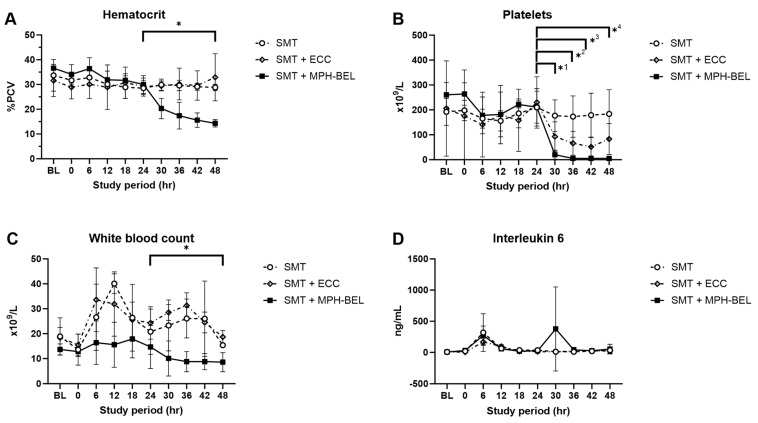
CBC results as well as Interleukin 6 values during the treatments. (**A**) Hematocrit, (**B**) Platelets, (**C**) White blood count, (**D**) Interleukin 6 values; SMT: standard medical therapy, SMT + ECC (extracorporeal circuit without BEL graft), SMT + MPH-BEL (bioengineered livers containing mixed populations of porcine hepatocytes and human endothelial cells). (**A**) * Significant difference (*p* < 0.05) of hematocrit in the comparison of SMT and SMT + MPH-BEL at T24 and T48; (**B**) *^1,^*^2,^*^3,^*^4^ Significant difference (Student’s *t*-test; *p* < 0.05) of platelet count level in the comparison of SMT and SMT + MPH-BEL at 24 h after 85% liver resection (T24) vs. 30 h after 85% liver resection (T30); 24 h after 85% liver resection (T24) vs. 36 h after 85% liver resection (T36); 24 h after 85% liver resection (T24) vs. 42 h after 85% liver resection (T42) and 24 h after 85% liver resection (T24) vs. 48 h after 85% liver resection (T48). (**C**) * Significant difference (Student’s *t*-test; *p* < 0.05) of white blood count in the comparison of SMT and SMT + MPH-BEL at 42 h (T42) and 48 h (T48) after 85% liver resection.

**Figure 8 biomedicines-12-01272-f008:**
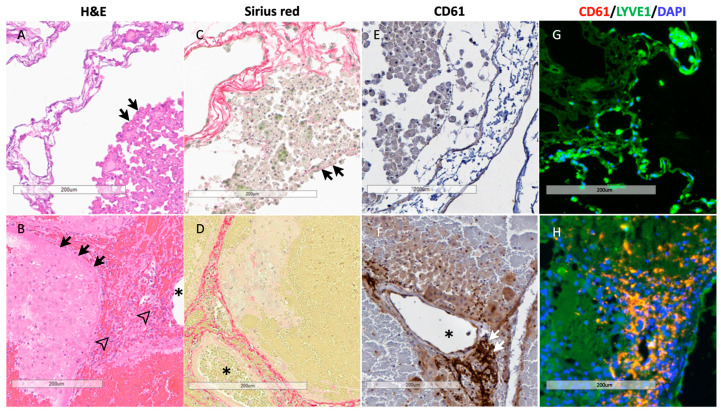
Representative histological slides of the PPLC-BEL following in vitro perfusion and following the PRLF treatment. (**A**,**B**): hematoxylin-eosin staining, successful recellularization with large numbers of hepatocytes (black arrows); post-PRLF treatment there is significant immune cell infiltration (framed arrowheads) in the periportal area (asterisk: portal vein); (**C**,**D**): Sirius red staining; preserved extracellular matrix in the periportal area (asterisk: portal vein) both after recellularization and after treatment, (**A**): fine extracellular matrix within hepatocyte aggregates; (**E**,**F**): CD 61 immunohistochemical staining; significant sequestration of platelets in the periportal area (asterisk: portal vein) after treatment; (**G**,**H**): immunofluorescence staining for CD 61, LYVE1 and DAPI; (**A**): successful endothelialization with HUVECs; (**B**): preserved endothelialization after treatment (continuous endothelium in lower right corner of image), significant immune cell infiltration and platelet sequestration in the periportal area.

**Figure 9 biomedicines-12-01272-f009:**
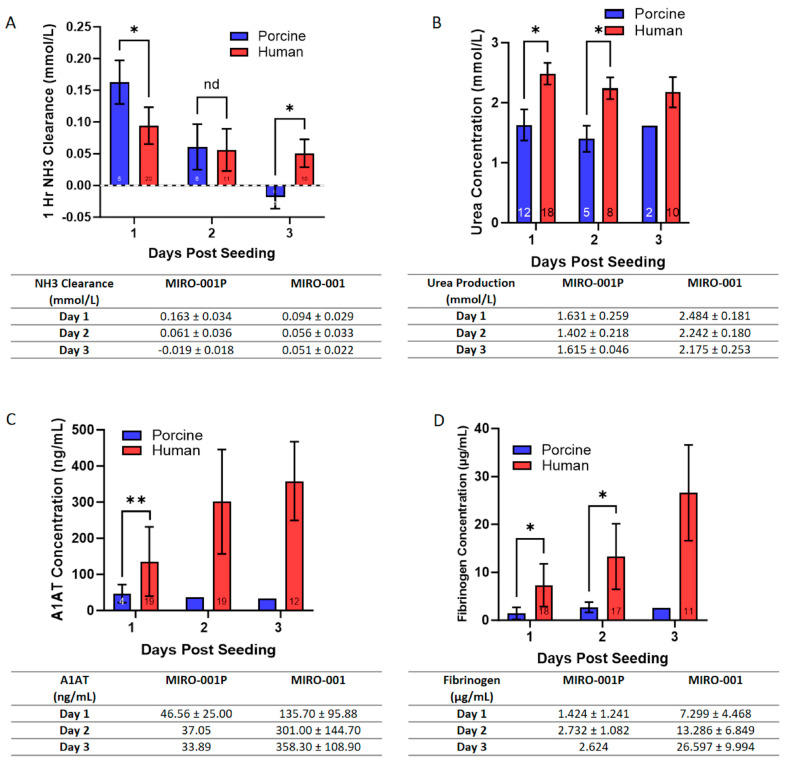
In vitro function of the MPH-BEL and the FH-BEL in the three-day long-term perfusion study. The function of the grafts was compared with regards to the detoxification and synthetic function (**A**–**D**) of the grafts. *: *p* < 0.05 using Student’s *t*-test; **: *p* < 0.01 using Student’s *t*-test; nd: no difference.

**Table 1 biomedicines-12-01272-t001:** Animal characteristics and operative variables. The *p*-value shows the statistical difference between the three individual groups by using the Kruskal–Wallis test. SMT: standard medical therapy, ECC: extracorporeal circuit without BEL graft, MPH-BEL: Bioengineered livers containing mixed populations of porcine hepatocytes and human endothelial cells.

	SMT	SMT + ECC	SMT + MPH-BEL	*p*-Value
Animal weight (kg ± SD)	31.4 ± 5.3	34.8 ± 2.3	32.8 ± 4.9	0.326
Pre-hepatectomy volume (mL ± SD)	941 ± 124	1003 ± 130	1020 ± 241	0.760
Post-hepatectomy volume (mL ± SD)	109 ± 32	133 ± 32	126 ± 38	0.718
Percent resection (%)	88.0	86.8	87.5	0.985
Blood loss during resection (mL ± SD)	254 ± 102	153.3 ± 50	193.8 ± 71	0.290
Intraoperative fluid (mL ± SD)	1242 ± 448	843 ± 266	681 ± 224	0.191

## Data Availability

Dataset available on request from the authors.

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
