# Peer review of "First Application of a Mixed Porcine–Human Repopulated Bioengineered Liver in a Preclinical Model of Post-Resection Liver Failure"

_biomedicines, 2024, doi:10.3390/biomedicines12061272_

Round 1

Reviewer 1 Report

Comments and Suggestions for Authors

1.It is better to describe the statistical analysis.

2.Albumin function is an important index, if the conditions permit in this study, it will better to exmaine the level of albumin concentrations in three groups.

3.The result section, line 378, the MPH-BEL originally written as the MPH-Bell.

4.To assess the function of MPH-BEL, just analysis the different time points in the respective study group, maybe it would be better ti compare between the groups.

5.In vitro experiment, the research group tested the fibrinogen concentration, the level is higher in FH-BEL graft, it has been verified that lower fibrinogen level was associated with bleeding in liver transplantation by Carrier FM et al., 2023.

6.Line 513, the FH-BEL originally written as the TH-BEL.

Author Response

Dear Reviewer, 

Thank you for sending the very helpful comments on our submitted manuscript. We have addressed all of your questions and modified the manuscript accordingly. 

Thank you very much 

Philipp Felgendreff

Reviewer 2 Report

Comments and Suggestions for Authors

The present study describes the effects of a mixed porcine-human repopulated bioengineered liver in a model of post-resection acute liver failure in pigs. Although the study is interesting and offers some new possibilities for further translational studies related to the new possibilites in the treatment of liver failure, some important issues must be addressed.

The Introduction section should be shortened.

The authors should present the schematic presentation of the extracorporeal BEL in the pig used in the present study in Figure 3.

The authors should not use t test for the estimation of the significance of the difference. There are three groups in the study, so ANOVA or adequate non-parametric test for multiple comparisons should be used.

Please explain the p values presented in the Table 1. What do they represent, the difference between which groups? Although authors describe the differences between groups, no significance is marked in figures.  Why did the authors decide to determine IL-6? There are also other cytokines that reflect the inflammation.

The authors claim that "The results of the study show that MPH-BEL treatment leads to improved detoxification and synthetic function as indicated by extended time of both the peak ammonia level and peak ICP results as well as accelerated INR normalization in the PRLF model.” However, the results do not indicate any significant difference in ammonia level. The authors describe the difference in the Results section, but figures do not support this finding. Please check once again all results with an appropriate statistical method.

Comments on the Quality of English Language

English is very good, only minor typographical errors are present. 

Author Response

Dear Reviewer, 

Thank you for sending the very helpful comments on our submitted manuscript. We have addressed all of your questions and modified the manuscript accordingly. Please see attached our point-to-point response. 

Thank you very much 

Reviewer 3 Report

Comments and Suggestions for Authors

In this study, Felgendreff and colleagues report the use of a human bioengineered liver (MPH-BEL) in a preclinical large animal study (post-resection liver failure, PRLF) demonstrating the potential utility of this system in case of acute liver failure (ALF).

The study is well designed, properly explained and the results are convincing and interesting. 

Since the causes of ALF are different, do the authors believe they can obtain the same results with all patients once the system is optimized or do they expect differences caused by the etiology of ALF? For example, they used the PRLF model. Would the authors expect the same results following ALF induced by intravenous injection of D-galactosamine (D-gal) (Cell Res 26, 206–216 (2016). https://doi.org/10.1038/cr.2016.6)

To be more precise/accurate, they should mention other bridging methods used in ALF, such as hepatocyte transplantation, in the introduction or during discussion, explaining the advantages of BEL over these alternative approaches.

Author Response

(The authors gave the same response as above.)

Round 2

Reviewer 2 Report

Comments and Suggestions for Authors

The authors have significantly improved the manuscript. My suggestion is that Fig. 3 should be deleted, because there is no reason to present extracorporeal BEL treatment in humans in Materials and Methods section, since no results in this study are obtained in human subjects. The abbreviations T24, T48, etc. should be explained in the Figure legends. Which statistical test was performed for each analysis? In my opinion ANOVA or Kruskall-Wallis test may be used for all analyses in the study, because there are three groups in the study. Pairwise comparisons are made according to post hoc tests.

Comments on the Quality of English Language

English is very good with some minor errors.

Author Response

Dear Reviewer, 

Thank you very much for reviewing the manuscript and for your very helpful questions. Please see the attached point-to-point letter and the updated manuscript. 

Thanks 

Philipp
